# Nonlinear Optical Properties from Engineered 2D Materials

**DOI:** 10.3390/molecules28186737

**Published:** 2023-09-21

**Authors:** Jia Shi, Shifeng Feng, Peng He, Yulan Fu, Xinping Zhang

**Affiliations:** 1Institute of Information Photonics Technology, Faculty of Science, Beijing University of Technology, Beijing 100124, China; fengshifeng@emails.bjut.edu.cn (S.F.); fuyl@bjut.edu.cn (Y.F.); zhangxinping@bjut.edu.cn (X.Z.); 2Department of Chemistry, National University of Singapore, 3 Science Drive 3, Singapore 117543, Singapore; hepeng07@nus.edu.sg

**Keywords:** nonlinear optics, 2D materials, SHG, THG, 2PPL, modulation

## Abstract

Two-dimensional (2D) materials with atomic thickness, tunable light-matter interaction, and significant nonlinear susceptibility are emerging as potential candidates for new-generation optoelectronic devices. In this review, we briefly cover the recent research development of typical nonlinear optic (NLO) processes including second harmonic generation (SHG), third harmonic generation (THG), as well as two-photon photoluminescence (2PPL) of 2D materials. Nonlinear light-matter interaction in atomically thin 2D materials is important for both fundamental research and future optoelectronic devices. The NLO performance of 2D materials can be greatly modulated with methods such as carrier injection tuning, strain tuning, artificially stacking, as well as plasmonic resonant enhancement. This review will discuss various nonlinear optical processes and corresponding tuning methods and propose its potential NLO application of 2D materials.

## 1. Introduction to Nonlinear Optics in 2D Materials

As a particular and interesting branch of modern optics, nonlinear optics mainly focus on the study of interaction between intense laser light and matter [1,2,3]. Shortly after the invention of the ruby laser in 1961, Franken et al. used a ruby laser with a wavelength of 694 nm to pass through a quartz crystal to generate ultraviolet light at 347 nm [4,5]. This is the earliest observed optical second harmonic generation (SHG) phenomenon, marking the birth of nonlinear optics. Nonlinear optics studies the phenomenon originating from the interaction between an intense laser with matter and has been widely used in laser generation, quantum optics, quantum computing, et al. [6,7,8,9,10]. Ordinary light sources with low intensity, such as sunlight and fluorescent lamps, cannot trigger the NLO responses, giving linear optical responses that obey superposition principles. Meanwhile, the intensity of a focused laser can reach 10^15^ to 10^22^ W/cm^2^, which means the electric field of the laser light source becomes comparable with that of the interatomic electric field (10^5^ to 10^8^ V/m) [11] and satisfies the requirement for a nonlinear response.

However, there are still many challenges in NLO applications, including high power loss [12], intricate nonlinear effects [13], and limited material response [14]. To solve these problems, researchers design high-performance platforms for integrated nonlinear optics to reduce the power loss [15], realize a large enhanced nonlinear response through resonant photonic nanostructures [16], and explore new materials with a high nonlinear coefficient [17]. Simultaneously, there are great challenges in integrating traditional nonlinear optical materials with other material platforms for on-chip applications [18,19]. Therefore, for future photonics and optoelectronics, it is very important to discover new materials with a large NLO response to meet the upcoming challenges through chip integration. These new materials are expected to provide innovative ways to enable new devices with new functions, high performance, and new device properties (e.g., reduced complexity, size, and cost, as well as a high nonlinear coefficient). 

NLO materials include organic and inorganic materials. The most common organic NOL materials include 4-*N*,*N*-dimethylamino-4′-*N*′-methyl-stilbazolium tosylate (DAST) crystal [20], chromophores [21,22,23], et al. Organic NLO materials show excellent properties, such as a high electro-optical coefficient, easy-to-produce significant nonlinear polarization, fast response rate, high bandwidth, and good processability [24]. The disadvantage is that the carrier mobility of organic materials is generally low and the carrier recombination rate is low, which limits the modulation rate of organic terahertz modulation devices to a certain extent [24]. In this review, we mainly focus more on the nonlinear optical properties and control methods related to inorganic materials.

In recent years, 2D materials attracted more and more attention from researchers since the atomic-layer-thick graphene was successfully obtained with mechanical exfoliation [25]. Compared to their 3D counterparts, the atomically thin thickness of 2D materials exhibits unique electronic and optical properties due to reduced dielectric shielding and enhanced Coulomb interactions [26,27,28,29,30,31,32]. With the advantages of an ultra-broadband absorption and large nonlinear optical response of 2D materials, they can be widely used in numerous nonlinear applications. For example, optical communication [33], frequency conversion [34,35], imaging and characterization [36,37], electro-optic modulators [38,39], mode-locking [40,41,42], and Q-switching [43]. Investigation and modification of the NLO properties of 2D materials facilitate the evolution of optical and optoelectronic devices.

In this review, we survey recent progress of NLO properties including SHG, THG, and 2PPL in 2D materials, and discuss effective methods for the realization of dynamic control of NLO properties. We first briefly describe the physical mechanisms of the nonlinear optical process of SHG, THG, and 2PPL and the phase-matching conditions in materials. Then, we mainly focus on the nonlinear optics characterization applied in 2D materials, i.e., layer number, crystal orientation, grain boundary, and probing excitonic states. Furthermore, we discuss different methods such as carrier injection, strain, artificially stacking, as well as plasmonic enhancement control to realize the dynamic control of nonlinear optical signals in 2D materials. Finally, we summarize the advantages and disadvantages of different control methods, and present a perspective of potential research topics.

### 1.1. Second Harmonic Generation (SHG)

For the nonlinear process, the light-induced polarization $\tilde{P}\left(t\right)$ is a power series of electric field $\tilde{E}\left(t\right)$, which can be described mathematically with the following equations [44,45]:(1)$$\tilde{P}\left(t\right)=\epsilon_{0}{[\chi }^{\left(1\right)}\tilde{E}\left(t\right)+\chi^{\left(2\right)}{\tilde{E}\left(t\right)}^{2}+\chi^{\left(3\right)}{\tilde{E}\left(t\right)}^{3}+\cdots ]\;\equiv {\tilde{P}}^{\left(1\right)}\left(t\right)+{\tilde{P}}^{\left(2\right)}\left(t\right)+{\tilde{P}}^{\left(3\right)}\left(t\right)+\cdots$$
where $\epsilon_{0}$ represents the vacuum permittivity and $\chi^{\left(1\right)}$ is the linear susceptibility, which plays a role in the linear process such as absorption. $\chi^{\left(2\right)}$ is the second-order NLO susceptibility, which determines processes such as the sum and difference frequency generation and electro-optic rectification. $\chi^{\left(3\right)}$, the third-order NLO susceptibility, is related to the third-order nonlinear effect, such as the third harmonic generation (THG) and two-photon photoluminescence (2PPL), four-wave mixing, Kerr lens effect, self-focusing, and white light generation with self-phase modulation [46]. The relationship between different orders of nonlinear susceptibilities and their applications is shown in Figure 1. ${\tilde{P}}^{\left(1\right)}\left(t\right)\;$ and ${\tilde{P}}^{\left(n\right)}\left(t\right)$ (*n* > 1) are the induced linear polarization and nonlinear polarization, respectively. 

Sum frequency generation occurs when two input photons with the frequency of $\omega_{1}$ and $\omega_{2}$ are combined, generating one photon with frequency $\omega_{3}$ due to the second-order nonlinear polarization. This is a parametric process that obeys the energy conservation, i.e., $\hbar \omega_{1}+\hbar \omega_{2}=\hbar \omega_{3}$. SHG is a special case of sum frequency generation, where only one input laser exists. Hence $\omega_{1}$ = $\omega_{2}$ and $\omega_{3}$ = 2$\omega_{1}$. The transition energy level diagram of SHG is represented in Figure 2; here, the $|\left.\phi \right\rangle$ represents the ground state, and the $|\left.\varphi_{v1}\right\rangle$ and the $|\left.\varphi_{v2}\right\rangle$ denotes two virtual states.

#### 1.1.1. Time and Spatial Inversion Symmetry

The second harmonic signal will vanish for the medium with inversion symmetry (as demonstrated below). When E changes to the reverse direction, i.e., $\tilde{E}\left(t\right)=-\tilde{E}\left(t\right)$, the polarization $\tilde{P}\left(t\right)$ will flip the sign and become $-\tilde{P}\left(t\right)$ accordingly: (2)$$-\tilde{P}\left(t\right)=\chi^{\left(2\right)}(-{\tilde{E}\left(t\right))}^{2}$$

According to Neumann’s principle, which indicates that the susceptibility tensor must be invariant by all the symmetry operators of the medium, the $\chi^{\left(2\right)}$ should be unchanged. Thus, combine $\tilde{P}\left(t\right)=$ $\chi^{\left(2\right)}({\tilde{E}\left(t\right))}^{2}$ and $-\tilde{P}\left(t\right)=$ $\chi^{\left(2\right)}(-{\tilde{E}\left(t\right))}^{2}$, and we obtain $\chi^{\left(2\right)}$ = 0. An expansion is that all the even-order susceptibility must be 0 for the medium with inversion symmetry.

#### 1.1.2. Phase-Matching Condition

The phase-matching condition should be achieved in order to achieve high conversion efficiency for nonlinear bulk crystal, which means Δk should be equal to zero ($\Delta k=k_{1}+k_{2}-k_{3}$) [48,49]. The momentum conservation determines this equation. The momentum of foundational frequency and the frequency doubling light can be described with the momentum conservation:(3)$$P_{2\omega }=\frac{n_{2\omega }\hbar 2\omega }{c_{0}}\;\mathrm{and}\;P_{\omega }=\frac{n_{\omega }\hbar \omega }{c_{0}}$$
where the ℏ is the reduced Planck constant; $c_{0}$ is the speed of light in the vacuum; ω is the angular frequency; and $n_{2\omega }$ and $n_{\omega }$ are the refractive indicies of the doubling frequency and the incident frequency, respectively. Due to the dispersion, generally, the refractive index of 2ω is larger than ω, making the condition of $P_{2\omega }$ = $P_{\omega }$ hard to fulfil. The phase matching can be achieved with birefringence.

### 1.2. Third-Order NLO Processes

This section briefly introduces two typical third NLO processes relevant to the research work on THG and 2PPL presented in this thesis. THG is similar to SHG, where three photons with the same energy $\hbar \omega_{1}$ are absorbed simultaneously, followed by generating a new photon with the energy of 3$\hbar \omega_{1}$. Figure 3a depicts the principle of THG. Unlike the $\chi^{\left(2\right)}$ that will vanish in crystal with inversion symmetry, $\chi^{\left(3\right)}$ always exists, independent of the crystal symmetry. However, $\chi^{\left(3\right)}$ is generally orders of magnitude weaker than $\chi^{\left(2\right)}$ for frequency conversion crystal [50]. An explanation from the perspective of quantum mechanics is that the possibility of three photons being present simultaneously is much less than two [51,52].

2PPL is induced with the two-photon absorption (TPA), an NLO absorption process involving simultaneous absorption of two photons and the transition from the ground state to the excited state [53]. After that, one photon with fixed energy is emitted, and the semiconductor decays back to the ground state, as shown in Figure 3b. Due to almost the same photoluminescent process, the emission spectra of 2PPL should generally possess a similar shape, peak position, and lifetime compared to the PL spectra [54,55,56,57,58].

It is easy to confuse the 2PPL with SHG, as 2PPL and SHG processes have some similarities, including the quadratic dependence of emission intensity on the excitation intensity. However, they are different: the emission of SHG is from a virtue state, whereas that of 2PPL originates from the real states. In other words, SHG is parametric while 2PPL is nonparametric, which involves absorption and energy relaxation. 

Finally, it is worth noting that one-photon and two-photon processes obey different selection rules. As the optical section rule relies on the final state symmetry, transitions of one-photon excitation are only allowed for final states with even parity (such as the s state), while the two-photon excitations can reach final states with odd parity (such as the p state). Thus, unlike linear optics, information about dark excitons can be probed from the wavelength-dependent 2PPL and SHG. 

## 2. Nonlinear Optics Characterization of 2D Materials

Two-dimensional materials with atomic layer thickness have large nonlinear optical coefficients and have attracted more and more attention. At the same time, since the nonlinear optical signals of 2D materials including SHG, THG, and 2PPL signals are very sensitive to the thickness, crystal phase, crystal axis orientation, grain boundary, and excitonic state of the material, nonlinear optics (NLO) spectroscopy can serve as a rapid, nondestructive, and accurate characterization method for 2D materials.

### 2.1. Determining the Layer Number 

Among 2D materials, transition metal chalcogenides (TMDCs) such as MoS_2_, WS_2_, WSe_2_, hBN, etc., have become the most popular research objects because of their excellent charge-carrier mobility, high nonlinear coefficients, and high switching ratio [59,60,61]. TMDC materials are relatively common and stable in nature mainly in 1T, 2H, and 3R crystal phases [62,63,64]. As we discussed above, SHG will vanish for the medium with inversion symmetry [65]. Xiang Zhang et al. reported an SHG generated from the 2H and 3R crystal phase of MoS_2_ and discussed the relationship between SHG intensity and thickness in thin-layer (1~6L) MoS_2_ [66]. As shown in Figure 4a, the odd layers of the 2H phase retain a net nonlinear dipole with noncentral inversion symmetry, whereas centrosymmetric even layers exhibit a zero second-order nonlinear coefficient. Therefore, the SHG intensity oscillates with the layer number in the 2H thin layer as exhibited in Figure 4c. Contrary to 2H-phase crystal, 3R MoS_2_ maintains broken inversion symmetry from the monolayer to bulk, and the layers retain the same orientation but shift along the in-plane direction, and leads to parallel in-plane nonlinear dipoles as shown in Figure 4b. When the thickness of the 3R MoS_2_ is less than half of the coherence length (L) between the fundamental incident laser and SH signal [67], the SHG intensity of 3R MoS_2_ (for 1–5 layers) is proportional to the square of its thickness as can be seen from Figure 4c. If the thickness of 3R MoS_2_ is larger than the coherent length, the optical path difference (OPD) between the upper and lower surface of thicker 3R MoS_2_ needs to be considered. A model including bulk nonlinear coefficient contribution and interface interaction (MoS_2_–air/MoS_2_–substrate interfaces) is proposed by Xinfeng liu et al. [68]. The distribution of SHG intensity oscillations with thickness can be well explained using this model (Figure 4d). 

Unlike SHG, THG can exist in structures with central inversion symmetry. José C. V. Gomes et al. studied the dependency between THG and the number of layers in WSe_2_ [69]. The results indicate that for thin-layered WSe_2_ (1~9L), the THG intensity is proportional to the square of the layer number (N^2^) as shown in Figure 4e. This quadratic scaling with thickness serves as direct evidence that each layer contributed independently to the overall THG, since the third-harmonic susceptibility of the *N* layer is proportional to that of the single layer: $\left|\chi_{N}^{\left(3\right)}\right|=N\times \left|\chi_{s}^{\left(3\right)}\right|$.

Similar to SHG, the influence of interference effects needs to be considered for dependence of the THG intensity on the thickness in thick-layered 2D materials. Vladimir O. Bessonov et al. reported THG intensity versus thickness on thicker-layered hexagonal boron nitride (hBN) and explained that the THG signal generated with the surface layer interferes with the generation of deeper layers, leading to a decrease in THG intensity [70]. In addition, the second interface (hBN/substrate) will affect the location of maxima and minima in the thickness dependence of THG (Figure 4f). Since SHG relies on the broken inversion symmetry, it can be used to judge parity layers in 2H-phase 2D materials. The distribution relationship between the SHG/THG signal intensity of 3R-phase/all-phase 2D materials and the thickness can be inferred according to the model considering the interface interference effect.

### 2.2. Crystal Orientation Identification and Mapping the Grain Boundary

Since crystal orientation and grain boundaries play a major role in the physical properties of two-dimensional layered materials, it is important to find efficient characterization methods to visualize them. Compared with traditional angle-resolved Raman spectroscopy [71,72], X-ray diffraction (XRD) [73,74], high-resolution transmission electron microscopy (HR-TEM) [75,76], and other characterization methods, the advantages of a high speed and nondestructive characterization have made nonlinear optical spectroscopy a powerful method for determining the orientation of the crystal axis and mapping the grain boundary. 

Because the effective second-order nonlinear coefficients $d_{eff}=1/2\chi^{\left(2\right)}$ are highly correlated with structural symmetry, angle-resolved SHG can be used as a fast and nondestructive characterization method for the crystallographic orientation of 2D materials. The polarization direction of the SHG depends on the direction of the incident electric field and $d_{eff}$ of the materials, which can be expressed with the following function [37]:(4)$$E\left(2\omega \right)\cdot {\hat{e}}_{2\omega }=C{\hat{e}}_{2\omega }\cdot \chi^{\left(2\right)}:{\hat{e}}_{\omega }{\hat{e}}_{\omega }$$
(5)$$I_{SHG}={\left|{\hat{e}}_{2\omega }\cdot d_{eff}\cdot {{\hat{e}}_{\omega }}^{2}\right|}^{2}$$
where ${\hat{e}}_{\omega }$ and ${\hat{e}}_{2\omega }$ represent the direction of the incident and outgoing electric field, C is the scaling factor that includes the local electric field factor. In the 2H-phase 2D materials (like hBN, MoS_2_, etc.), the odd-numbered layer structure belongs to the D_3h_ point group. Through the simplification of crystal symmetry, $d_{eff}$ satisfied the matrix as follows:(6)$$d_{eff}=\left[\begin{array}{c} 0\\ d_{21}\\ 0 \end{array}\begin{array}{c} 0\\ d_{22}\\ 0 \end{array}\begin{array}{c} 0\\ 0\\ 0 \end{array}\begin{array}{c} 0\\ 0\\ 0 \end{array}\begin{array}{c} 0\\ 0\\ 0 \end{array}\begin{array}{c} d_{16}\\ 0\\ 0 \end{array}\right]$$

Among them (*d*_21_ = *d*_16_ = −*d*_22_), if the incident light is perpendicular to the sample and the incident electric field is projected onto the armchair direction of the sample as shown in Figure 5a, the parallel and perpendicular SHG of 2H-phase TMDCs can be written as
$$I_{∥}=I_{0}{cos}^{2}\left(3\theta \right)$$
(7)$$I_{⊥}=I_{0}{sin}^{2}\left(3\theta \right)$$

Here, θ represents the azimuth angle of the material, and $I_{0}$ is the maximum intensity of the SHG. Since θ=0° corresponds to a direction where the polarization of the pump beam is along the armchair direction, the SHG intensity reaches its largest state. Tony F. Heinz and José C. V. Gomes et al. found that the SHG intensity polar plot of odd-layered hBN, MoS_2_, and WSe_2_ exhibits a six-lobed shape, which corresponds to the D_3h_ symmetry as shown in the right side of Figure 5a and the left side of Figure 5b, respectively [69,77]. In contrast, José C. V. Gomes et al. carried out the polarization-dependent THG measurement on monolayer WSe_2_ and reported that the intrinsic THG response is independent of the incident polarization (Figure 5b) [69]. Kaihui Liu et al. also performed a similar experiment and monitored the pattern evolution in monolayer WS_2_ [78]; the results confirmed that the generated THG satisfied the relationship with the polarized incident laser, $I_{∥}=I_{0}$, $I_{⊥}=0$, which means that THG is polarization-independent. Polarization-resolved nonlinear spectroscopy can identify the crystal axis orientation, while a THG image can realize the visualization of grain boundaries. Lasse Karvonen et al. carried out multiphoton (SHG and THG) characterization studies by scanning a laser beam over the MoS_2_ flakes (Figure 5c) with two galvo mirrors; SHG and THG images obtained without an analyser in front of the detector are shown in Figure 5c,f [79]. In the total-SHG image, only GB1 appears as dark due to the destructive interference of the SHG fields from the neighbouring grains [80]. Figure 5e presents results for polarized SHG imaging (parallel-polarized excitation and detection); only the area above GB1 is visible, suggesting a large difference between the upper (A1 and A2) and lower (B1 and B2) parts, which indicates that the polarized SHG image can resolve the crystal orientations. The contrast difference between neighbouring grains is due to the anisotropic polarization pattern of SHG [60], and the occurrence of destructive interference depends on whether the edges of the neighbouring grains are of the same type and the incident light polarization is perpendicular to the bisecting direction of the grain edges [81]. Contrasting the SHG image, all the GBs (GB1-GB4) can be clearly seen from the THG image (Figure 5f). Since THG does not depend on the parity of the layer number and exhibits no polarization dependence on the crystallographic axis angle, a THG image can serve as a rapid and sensitive visualization method of grain boundaries.

**Figure 5 molecules-28-06737-f005:**
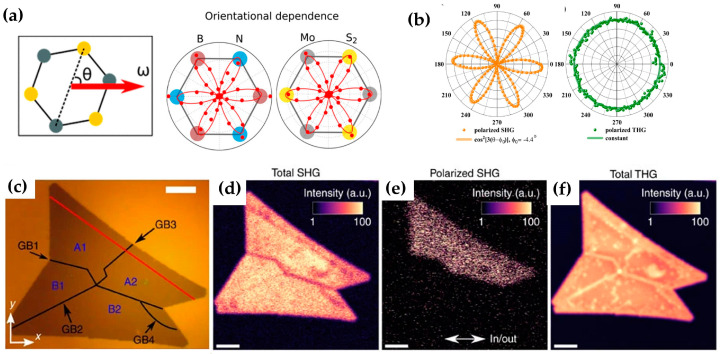
(**a**) Left: schematic diagram of the atomic structure of TMDC material, where θ represents the azimuth angle of the material. The angle θ=0° corresponds to a direction where the polarization of the pump beam is along the armchair direction. Right: schematic view of SH intensity polar plot of monolayer hBN and MoS_2_; (**b**) polarization-dependent SHG and THG of WSe_2_; (**c**) optical image of MoS_2_ with marked grains (A1, A2, B1, and B2) and grain boundaries (GB1, GB2, GB3, and GB4); (**d**) experimental SHG mapping image of MoS_2_ without analyser, scale bars: 10 μm; (**e**) experimental SHG image with parallel polarized excitation and detection; (**f**) experimental THG mapping of MoS_2_ without analyser; panel (**a**) adapted with permission from Ref. [77], Copyright 2013, American Chemical Society; panel (**b**) adapted with permission from Ref. [69], Copyright 2018, American Chemical Society; panel (**c**–**f**) adapted with permission from Ref. [79], Copyright 2017, Nature communications.

### 2.3. Probing Excitonic State

Monolayer TMDs with atomic thickness usually exhibit a strong Coulomb interaction as a result of reduced dielectric screening effects. This leads to the formation of a series of discrete excitonic states near the band edge called “Rydberg” excitons [82,83,84]. As can be seen from Figure 6a, the “Rydberg” excitonic states below the quasiparticle bandgap generally exhibit electronic energy levels analogous to the hydrogen series labelled as 1s, 2p, 3p, and so on [85]. If the incident electromagnetic wave is in resonance with the energy of the exciton states, light–matter interaction will be significantly enhanced [86,87]. Whether it is the giant PL emission enhancement [88,89] or the nearly 100% light absorption near the exciton state [90], it proves the existence of the exciton resonance effect. Similarly, the exciton resonance effect in TMDs is also applicable to its nonlinear optical properties. For example, signals of SHG, 2PPL, and THG can be greatly enhanced by up to three orders of magnitude if the excitation energy is resonant with the corresponding excitonic state. Based on the symmetry selection rules of electronic transition, a one-photon excitation process can only reach states with even parity (dipole-allowed bright states) while two-photon transitions can reach odd-parity states with nonzero orbital angular momentum. This means that bright states like “ns” can be detected with optical reflectance spectroscopy [82,83] or photoluminescence (PL) excitation (PLE) spectroscopy [91]. The dark states like “np” and “nd” can be probed with two-photon photoluminescence excitation (2PPLE) spectroscopy [85]. Since the generation of an SHG signal is the combination of electric–dipole (ED) and magnetic–dipole (MD) optical transitions, SHG excitation spectroscopy (SHGE) can probe both even (“s”) and odd (“p”) parity states [92]. The generation of THG does not depend on the broken inversion symmetry, and its resonance excitonic energy is high. Therefore, it can not only detect bright “ns” states but also high-energy excitonic states like C and D excitons.

Based on the above principle, G. Wang et al. carried out an SHGE, PLE, as well as 2PPLE experiment on monolayer WSe_2_ at 4K as shown in Figure 6b and the excitation power was kept constant in the experiments [92]. According to the transition selection rule discussed above, the SHGE spectrum detects “1s”, “2s/2p”, and “3s/3p” excitonic states of the A exciton as well as “1s” and “2s/2p” excitonic states of the B exciton. However, the energy difference between the “s” and “p” states is too small to distinguish from the SHGE spectrum. Since 2PPLE and PLE can only detect dark “p” states and bright “s” states, respectively, combining them with SHGE can achieve a detailed identification of Rydberg exciton states. Ziliang Ye et al. also similarly probed the excitonic effects in monolayer WS2 and demonstrated that there is no significant shift in the excitation energy of either the “s” or the “p” states with different capping layers by measuring 2PPLE of monolayer WS2 with different dielectric capping layers, including water, immersion oil, and aluminium oxide [85]. Yadong Wang et al. demonstrated broadband THGE (≈1.79 to 3.10 eV) in monolayer MoS2 to explore prominent fingerprints of excitonic states including “s” series excitons and free-band electronic states [93]. As exhibited in Figure 6c, remarkable THG enhancement at certain photon energies has been observed and the effective third-order nonlinear susceptibility at different THG energies was calculated. By fitting the peak labelled as P1-P7 in Lorentzian functions and comparing that with the second-order differential of the reflectance spectrum, the authors successfully identified “1s” and “2s” of the A exciton as well as the 1s state of B, C, and D excitonic states with a higher energy level. 

Therefore, broadband and high-contrast nonlinear optical spectrums can serve as new characterization methods to uncover several relevant electronic states such as bright s series, dark excitons, as well as band-edge excitonic states. This nondestructive and fast method of determining the excitonic state facilitates our further study of the physical properties related to the exciton in 2D materials.

## 3. Tuning Methods

Nonlinear light-matter interaction in atomically thin TMD is important for both fundamental research and future optoelectronic devices. In recent years, researchers have tried different methods such as carrier injection, strain, artificially stacking, as well as plasmonic enhancement control to realize the dynamic control of nonlinear optical signals.

### 3.1. Carrier Injection Tuning

Xiong’s group demonstrated the emergence of an SHG signal in 2L WSe_2_ with plasmonic hot-electron injection as shown in Figure 7a [94]. The SHG is absent in 2L WSe2 due to the centrosymmetry; however, when the gold nanorod is coupled with 2L WSe2, the SHG signal of the bilayers changes from nothing to a presence as shown in Figure 7b. Under the excitation of an 850 nm pump laser, only the plasmonic metal can be excited, and the rise time around 120 fs and the decay lifetime around 1 ps are consistent with that of plasmonic hot-electron generation and transfer time. The mechanism involved was considered to be the broken inversion symmetry in 2L WSe2 with hot-electron injection and thus inducing an SHG. Similarly, Xiong’s group also demonstrated manipulation of the bond charges in a 2L WSe2 by applying a perpendicular electric field. They reported the charge-induced SHG (CHISHG) from the 2H-2L-WSe2-based back-gate field transistors, shown in Figure 7c–e [95]. Under the gate voltage of −40 V, the SHG intensity from bilayers was found to be 1000 times weaker than that from the monolayer. There is a charge (hole) accumulation in the sample and efficient SHG can be observed, as shown with the back-gate sweep of source-drain current Isd in Figure 7e. The authors explained the origination of SHG with a bond charge model: charge accumulation in the W metal planes leads to a nonuniform electric distribution, which breaks the symmetry of bilayers. In another work, Xiaodong Xu et al. reported the electrical control of SHG intensity regarding a monolayer WSe2 field-effect transistor [39]. They chose the electrostatic doping method induced with strong exciton charging effects in monolayer TMD materials. If the two-photon excitation energy is resonant with the A exciton as shown in Figure 7f, the resonant SHG can be tuned with electrostatic doping and exhibits a nearly four-fold intensity reduction and gate voltage (Vg) is swept from −80 to 80 V. If the two-photon excitation energy is 30 meV above or below the exciton resonance, Vg has no effect on the SHG (Figure 7g), supporting the idea that the SHG tuning originates from the modulation of oscillator strength at the exciton resonance.

### 3.2. Strain Tuning

Mechanical strain can deform the crystal lattice, thus enabling the modulation of SHG. For example, Mennel et al. found that strain can alter the polarization dependence of SHG for monolayer MoS_2_ [96]. Figure 8a,b show that under the 1% strain, the polarization pattern of SHG deforms along the uniaxial strain. Electric gating and charge injection can also be utilized to break the symmetry, and this fascinating property is only possible in TMDCs. In another work, Li et al. reported a heterostructure combining mechanically exfoliated MoS_2_ and TiO_2_ nanowires and realized the modulation of the second harmonic polarization of MoS_2_ through the spontaneous strain of different orientations, as shown in Figure 8c [97]. The MoS_2_/TiO_2_ heterojunction region is ten times stronger than the SHG of monolayer MoS_2_, while the region of pure TiO_2_ does not produce any signal. It proves that the enhancement of the second harmonic signal in this heterojunction region does not originate from titanium dioxide direct contribution, as shown in Figure 8d. Therefore, in layered 2D materials, since the second harmonic is very sensitive to strain, strain modulation of the second harmonic in 2D materials is very promising.

### 3.3. Artificially Stacking

SHG is highly sensitive to lattice symmetry and crystallographic axis orientation of the material. Hence, it can be modulated with artificial symmetry control, such as manipulating the adjacent layer alignment and applying external stimulation. Lin et al. stacked multiple triangular WS_2_ sheets in a concentric manner to form a pyramid, shown in Figure 9a [98]. With the number of layers superimposed, the SHG intensity will show a trend of decreasing oscillation similar to that of 2H. With more than 70 layers, the SHG intensity will significantly occur. The main reason for the enhancement is due to the edge SHG enhancement of the multilayer structure, which will eventually be 45 times stronger than that of a single layer, as shown in Figure 9b. Furthermore, Fan et al. reported a detailed SHG study of spiral WS_2_ multilayer nanosheets with a twisted angle of 5 degrees, as shown in Figure 9c [99]. In contrast to the diminishing oscillation of normal 2H-phase WS_2_, the SHG intensity of spiral WS_2_ increases with the layer number (see Figure 9d). This spiral WS_2_ with enhanced SHG signals in multilayers extends the NLO application of TMDCs. Hsu et al. in 2014 first started the investigation of SHG in TMDC heterostructures. They fabricated chemical vapor deposition (CVD)-grown homo-stacked MoS_2_/MoS_2_ and studied the SHG generation with different stacking angles, as shown in Figure 9e [100]. Their experiment demonstrated that SHG from the artificial bilayer is a coherent superposition of that from individual layers, with a phase difference determined with the stacking angle. For a bilayer with a 2H stacking order (θ = 60°), the SHG totally vanishes, and for a 3R stacking order AA (θ = 0°), the SHG is nearly three times stronger than that from the monolayer, shown in Figure 9f. In addition to stacking the same TMDC material, there is also the stacking of different materials to form a heterojunction structure to enhance the SHG and THG. Zhang et al. fabricated a typical monolayer WS_2_ (WSe_2_) and MoS_2_ (MoSe_2_) heterostructure, keeping its triangular domain shape, as shown in Figure 9g [101]. Figure 9h,i show the parallel-polarized SHG images of the lateral heterostructures of WSe_2_-MoSe_2_ and WS_2_-MoS_2_, respectively. The maximum SHG image of the lateral heterostructure of WS_2_-MoS_2_ was obtained at the position of θ = 0°.

### 3.4. Plasmonic Enhancement

There are two types of plasmons: localized surface plasmons and surface plasmons. Surface plasmons refer to the collective oscillation of free electrons on the metal surface when light is incident on the interface between the metal and the dielectric. When the frequency of the incident light is the same as the oscillation frequency of the electrons, the electromagnetic field will be confined on the metal surface and generate. The enhancement of the local electric field generated with plasmons can effectively improve the conversion efficiency of the second harmonic. Local surface plasmons can effectively confine light to the sub-wavelength region, thereby significantly enhancing the local electromagnetic field. Shi et al. reported a 400-fold SHG enhancement by coupling the monolayer WS_2_ to the 2D Ag nanogrooves_._ as shown in Figure 10a [102]. The resonant electric field inside the nanogroove was designed at 430 nm, overlapping with the C exciton resonance in WS_2_. In addition, the presence of grooves provides much stronger enhancement than that with the Ag plant, as shown in Figure 10b. Wang et al. combined the flexible substrate PDMS with a single layer of WS2 to design a surface plasmon structure [103], which achieved a second harmonic enhancement of more than three orders of magnitude, as shown in Figure 10c. A 7000-fold SHG enhancement at the channel was finally obtained by controlling the size of the strip structure, as shown in Figure 10d. Previously, SHG or THG was enhanced separately, and SHG and THG were not enhanced at the same time. Shi et al. designed a hybrid nanostructure of monolayer WS_2_ integrated with a plasmonic cavity as shown in Figure 10e [104]. Optimum 1000-, 3000-, and 3800-fold enhancement was achieved for two-photon photoluminescence, second harmonic generation, and third harmonic generation, respectively, in the optimized cavity structure, which is composed of the Au/SiO_2_ (50 nm/10 nm) substrate, monolayer WS_2_, PAH, and CTAB-coated single AuNH, as shown in Figure 10f. Meanwhile, results show that plasmon enhanced NLO spectroscopy could serve as a general method for probing high-order Rydberg excitonic states of 2D materials. With the maturity of the surface plasmon technology and the in-depth understanding of the localized surface plasmon resonance, it is possible to design a more refined structure that can better confine the incident electric field in the future to increase the SHG and THG.

## 4. Summary and Outlook

In this review, we summarize the research progress on typical nonlinear optical (NLO) processes in 2D materials, including SHG, THG, and 2PPL. It is worth noting that for 2D materials, not only the inorganic TMD materials mentioned in the review show interesting nonlinear optical properties but also organic 2D materials such as covalent organic frameworks (COFs), metal–organic frameworks (MOFs), and chromophore-conjugated nanomaterials exhibit attractive nonlinear optical properties, especially the giant two-photon absorption (2PA) [105,106,107]. For example, fullerene derived with chromophore-conjugated nanomaterials corresponds to 2D characters due to the interactive fullerenyl hydrophobic intermolecular interaction forces and exhibits large cross-sections of 2PA [108,109]. Compared to 2D inorganic materials, molecular chromophores are advantageous in larger cross-sections and tuning the 2PA properties with different functional groups; however, it is challenging to precisely control the spatial arrangement of the chromophores and the chromophore molecules usually exhibit poor orientation stability [24]. For inorganic 2D TMD materials, they have good stability and exhibit a high SHG and THG nonlinear coefficient. However, their 2PPL signals are still relatively limited due to their weak quantum efficiency and small cross-section. Therefore, appropriate enhancement and regulation methods are still urgent issues that researchers need to solve.

In addition, we demonstrate that the nonlinear optical spectroscopy can serve as a characterization method for 2D materials such as determining the layer number, crystal orientation identification, and mapping the grain boundary and probing excitonic state. By means of carrier injection tuning, strain tuning, artificial stacking, and surface plasmon enhancement, nonlinear optical signals in 2D materials can be effectively modulated. Table 1 shows the above kinds of regulation methods and modulation effects. 

The NLO feature has the characteristics of high-speed measurement, nondestructive measurement, large-area measurement, and multi-mode. Combining different tuning methods can have a large number of applications in optoelectronic devices like all-optical devices, photonic information communication, and even nonlinear optical valleytronics. For example, combining the plasmonic modulation and nonlinear optical signals can be applied in the application of nonlinear valleytronics in TMDC materials [110,111]. The use of strain tuning as well as excitonic resonant excitation can broaden the modulation depth and range and further improve the performance of the NLO response of TMDC [112]. 

Finally, for the future development direction, we can explore the nonlinear optical properties of new 2D materials and their heterostructures such as 2D magnetic materials and related control methods, and create highly stable and high-quality nonlinear optical integrated devices to broaden their application fields in semiconductor electronics, spin valley electronics, and other fields.

## Figures and Tables

**Figure 1 molecules-28-06737-f001:**
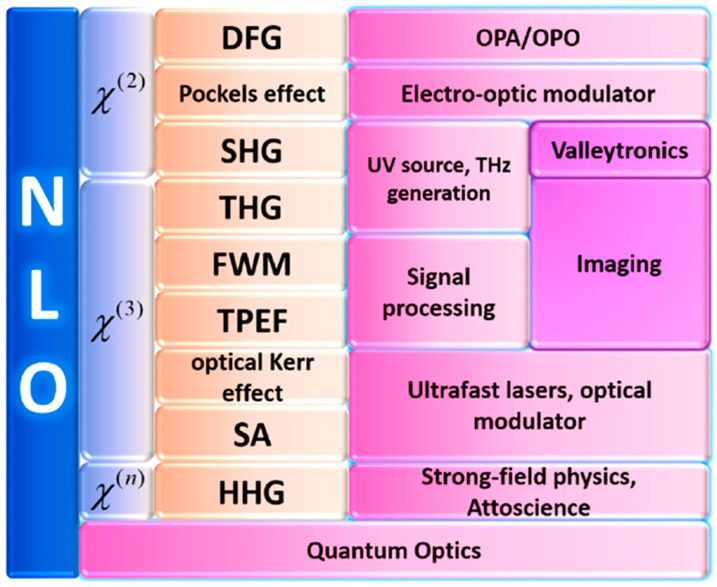
Summary of different orders of NLO susceptibilities and their applications, reprinted from reference [47].

**Figure 2 molecules-28-06737-f002:**
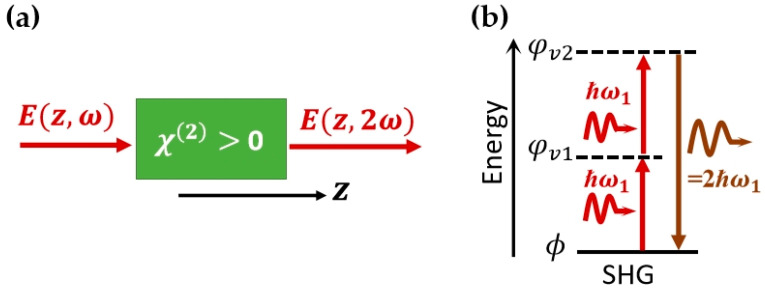
Schematic (**a**) and energy diagram (**b**) of the SHG process.

**Figure 3 molecules-28-06737-f003:**
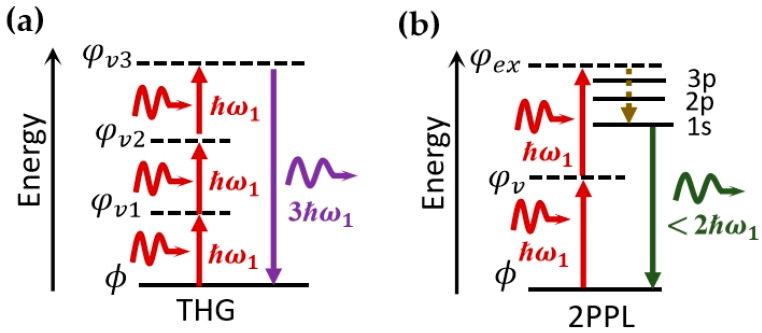
Energy diagrams of (**a**) THG and (**b**) 2PPL processes.

**Figure 4 molecules-28-06737-f004:**
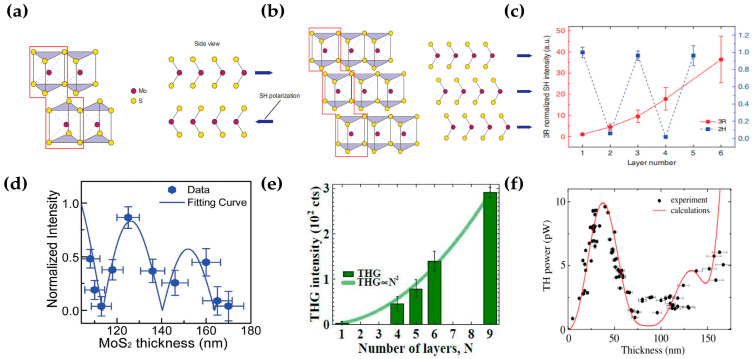
(**a**) Schematic of the crystal structure of 2H- and 3R-phase MoS_2_ (two-layer unit cell outlined in red) with Mo atoms in maroon and S atoms in yellow; (**c**) SH intensity of the 3R and 2H MoS_2_ normalized to the respective single-layer intensity at an SH energy of 1.81 eV; (**d**) THG peak intensity of WSe_2_ as a function of layer number *N* (*N* = 1 … 9); (**e**,**f**) thickness dependence of SHG and THG peak intensity for 3R MoS_2_ and 2H hBN, respectively; panel (**a**–**c**) adapted with permission from Ref. [66], Copyright 2014, American Chemical Society; panel (**d**) adapted with permission from Ref. [68], Copyright 2017, American Chemical Society; panel (**e**) adapted with permission from Ref. [69], Copyright 2018, Scientific Reports; panel (**f**) adapted with permission from Ref. [70], Copyright 2014, American Chemical Society.

**Figure 6 molecules-28-06737-f006:**
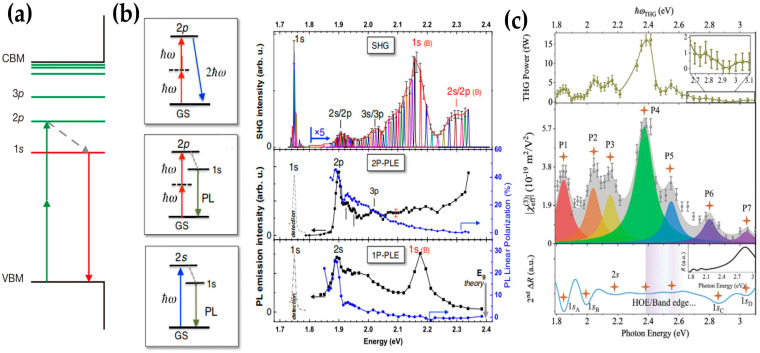
(**a**) Schematic diagram of two-photon luminescence (2PPL) process in monolayer TMDCs; (**b**) left side from top to bottom: schematic for SHG/2PPLE/PLE when incident photons are resonant with the 2p/2s state of the A exciton in monolayer WSe_2_. Right side from top to bottom: results of SHG/2PPLE/PLE spectroscopy at T = 4 K as a function of 2ℏω; (**c**) from top to bottom: THG-power/third-order effective nonlinear optical coefficient/second-order linear reflectance (2nd ΔR) as a function of resonant photon energy from monolayer MoS_2_; panel (**a**) adapted with permission from Ref. [85], Copyright 2014, Nature; panel (**b**) adapted with permission from Ref. [92], Copyright 2015, Physical Review Letters; panel (**c**) adapted with permission from Ref. [93], Copyright 2022, Advanced Materials.

**Figure 7 molecules-28-06737-f007:**
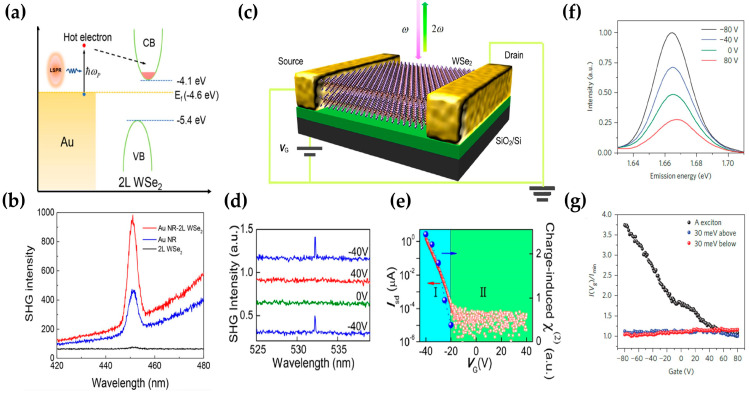
(**a**) Schematic view of plasmonic hot-electron transfer process in WSe_2_/Au-nanorod-coupled structure; (**b**) SHG intensity of gold nanorod array (Au NR), Au NR/bilayer WSe_2_, and bilayer WSe_2_, respectively; (**c**) schematic view of the WSe_2_ device configuration; (**d**) SHG spectra of WSe_2_ under different back gate; (**e**) I_sd_ curve of the bilayer WSe_2_ device as a function of V_G_; (**f**) SHG spectra of monolayer WSe_2_ where the two-photon excitation energy has resonance with the exciton at selected gate voltages; (**g**) normalized SHG peak intensity versus gate voltage when two-photon excitation energy is above, below, and resonant with the A exciton, respectively. Panel (**a**,**b**) adapted with permission from Ref. [94], Copyright 2018, American Chemical Society; panel (**c**–**e**) adapted with permission from Ref. [95], Copyright 2015, American Chemical Society; panel (**f**,**g**) adapted with permission from Ref. [39], Copyright 2015, Nature Nanotechnology.

**Figure 8 molecules-28-06737-f008:**
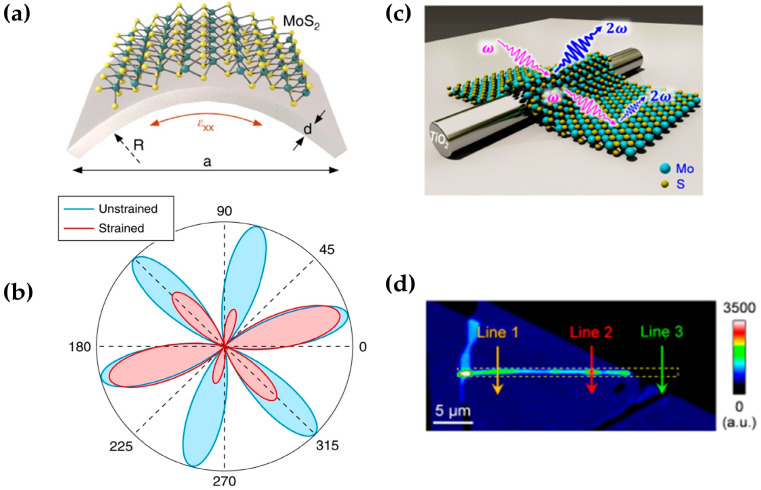
(**a**,**b**) Schematic showing the two-point bending method for applying uniaxial strain and the change of polarization-dependent SHG pattern under strain, reprinted from reference, with permission from Springer Nature; (**c**) image of TiO_2_/MoS_2_ heterojunction junction region; (**d**) characterization of second harmonic intensity imaging; panel (**a**,**b**) adapted with permission from Ref. [96], Copyright 2018, Nature Communications; panel (**c**,**d**) adapted with permission from Ref. [97], Copyright 2018, Nature Communications.

**Figure 9 molecules-28-06737-f009:**
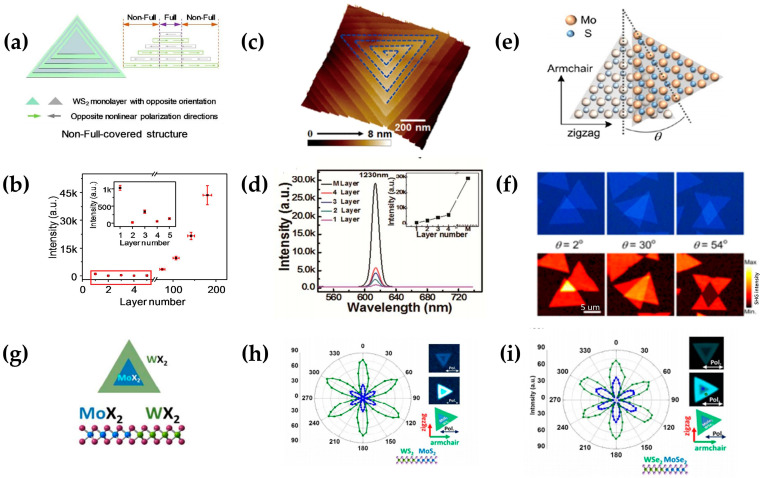
(**a**) Artificial synthesis of pyramidal WS_2_ nanostructures; (**b**) the dependence of the number of layers of pyramidal WS_2_ nanosheets on the strength of SHG; (**c**) optical image of the spiral WS_2_ nanosheet grown using CVD; (**d**) layer-dependent SHG spectra of the spiral WS_2_; (**e**) a schematic illustration of the artificially stacked bilayer MoS_2_ with arbitrary stacking angle θ; (**f**) optical microscopy images and SHG intensity mapping of bilayer MoS_2_ with different stacking orders; (**g**) heterostructure of TMD atomic layers; (**h**) polar plot of the parallel polarization SHG intensity of lateral heterostructure of WS_2_-MoS2 atomic layers; (**i**) polar plot of the parallel polarization SHG intensity of lateral heterostructure of WSe_2_-WSe_2_ atomic layers; panel (**a**,**b**) adapted with permission from Ref. [98], Copyright 2018, American Chemical Society; panel (**c**,**d**) adapted with permission from Ref. [99], Copyright 2017, American Chemical Society; panel (**e**,**f**) adapted with permission from Ref. [100], Copyright 2014, American Chemical Society; panel (**g**–**i**) adapted with permission from Ref. [101], Copyright 2015, American Chemical Society.

**Figure 10 molecules-28-06737-f010:**
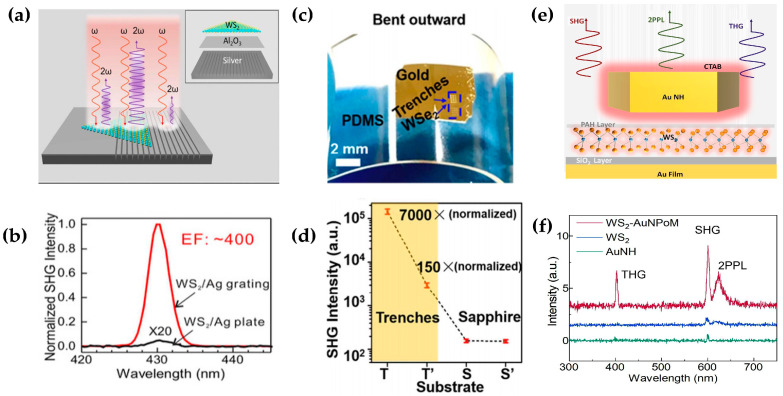
(**a**) Schematic illustration of Ag-WS_2_ hybrid metasurface structure, (**b**) SHG spectra of 1L WS_2_ on and off nanogroove cavity; (**c**) monolayer WSe_2_ on PDMS substrate, (**d**) PDMS surface plasmon structure enhances SHG by 400 times; (**e**) schematic side view of the designed WS_2_-AuNPoM hybrid structure. (**f**) NLO spectra of WS_2_-AuNPoM, WS_2_, and AuNH; panel (**a**,**b**) adapted with permission from Ref. [102], Copyright 2018, Photonics Reviews; panel (**c**,**d**) adapted with permission from Ref. [103], Copyright 2018, American Chemical Society; panel (**e**,**f**) adapted with permission from Ref. [104], Copyright 2022, American Chemical Society.

**Table 1 molecules-28-06737-t001:** Summary of different tuning methods.

Tuning Methods		Material	Regulatory Range
Carrier injection	Plasmonic hot electron	2LWSe_2_	Zero to nonzero [94]
	Charge accumulation	2L WSe_2_	Zero to nonzero [95]
	Electrical gating	1L WSe_2_	a factor of four [39]
strain tuning	uniaxial strain	MoS_2_	1.0% tensile strain [96]
	heterojunction strain	MoS_2_	10 times enhanced [97]
	triangle stack	WS_2_	45 times enhanced [98]
Artificially	spiral stack	WS_2_	hundreds of times 1 layer -M layer [99]
stacking	homojunction stacking	MoS_2_	4 times enhanced stacking angle 0–60° [100]
	heterojunction stacking	MoX_2_-WX_2_	6-fold symmetry [101]
	enhanced SHG	WS_2_	400 times enhanced [102]
Plasmonic	enhanced SHG	WSe_2_	7000 times enhanced [103]
Enhancement	enhanced SHG and THG	WS_2_	SHG 3000 times enhanced THG 3800 times enhanced [104]

## Data Availability

Not applicable.

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
