# Peer review of "Nonlinear Optical Properties from Engineered 2D Materials"

_molecules, 2023, doi:10.3390/molecules28186737_

Round 1

Reviewer 1 Report

In this paper, authors provide a comprehensive overview of recent advancements in the field of nonlinear optics (NLO) properties, specifically focusing on second harmonic generation (SHG), third harmonic generation (THG), and two-photon photoluminescence (2PPL) in 2D materials. Meanwhile, this review also discusses effective methods for achieving dynamic control of NLO properties. These include carrier injection, strain engineering, artificially stacking different layers, and utilizing plasmonic enhancement. The presented information could inspire further research efforts and advancements in this rapidly evolving field. However, some minor issues should be corrected and supplemented before acceptance. 

1.     In the “introduction” part, the authors mentioned specific challenges of nonlinear optics in two-dimensional materials, such as intricate nonlinear effects and limited material response. It is better to add some relevant literature to explain these challenges.

2.     In the “introduction” part line 39, the connection between current challenges and future goals is unclear, namely, what research is being done to address this problem and what problems are being solved to address these challenges.

3.     Figure 3f shows no scale bar in corresponding optical and SHG images.

4.     There are some errors with the format of the reference. For example, reference 2 lakes the page information, and reference 12 should be provided DOI. 

Some minor typos need to be doubly proof check

Reviewer 2 Report

The presented article is an interesting review regarding the design of 2D inorganic materials bearing nonlinear optical properties. I believe that the article is suitable for publication in the journal Molecules. However, there are some important comments that need to be corrected before this article can be published.

1. When reading the review, one gets the strong impression that the authors have made a one-sided emphasis when considering nonlinear optical materials. The authors should add a paragraph about the prospects, advantages and disadvantages of nonlinear optical materials based on organic compounds. For example, many research groups are developing nonlinear optical materials based on heterocyclic azines. It is strongly recommended to add the necessary references to the text of the review, for example, https://doi.org/10.1016/j.dyepig.2020.108509; https://doi.org/10.1016/j.dyepig.2020.108659; https://doi.org/10.1016/j.jphotochem.2020.112900; etc.

2. Some figures are too small to see them. Improve the dimensions of each image for a better see.

Round 2

Reviewer 2 Report

The authors have made an efficient revision on their work, which can be accepted in the current form.